# Rock Magnetism of Lapilli and Lava Flows from Cumbre Vieja Volcano, 2021 Eruption (La Palma, Canary Islands): Initial Reports

Josep M. Parés [1,*], Eva Vernet [2], Manuel Calvo-Rathert [2], Vicente Soler [3], María-Felicidad Bógalo [2] and Ana Álvaro [1]

1 Geochronology & Geology, CENIEH, Paseo Sierra de Atapuerca 3, 09002 Burgos, Spain; ana.alvaro@cenieh.es
2 Departamento de Física, EPS, Universidad de Burgos, Av. Cantabria, s/n, 09006 Burgos, Spain; evernet@ubu.es (E.V.); mcalvo@ubu.es (M.C.-R.); mfbogalo@ubu.es (M.-F.B.)
3 Instituto de Productos Naturales y Agrobiología (IPNA), Avda. Astrofísico Francisco Sánchez, 3, 38206 San Cristóbal de La Laguna, Spain; vsoler@ipna.csic.es
* Correspondence: josep.pares@cenieh.es

**Abstract:** We present initial rock magnetic results for both lava flows and lapilli produced by the 2021 eruption of the Cumbre Vieja, La Palma (Canary Islands). Samples were taken during the eruption to minimize early alteration and weathering of the rocks and tephra. Standard procedures included progressive alternating field and thermal demagnetization, hysteresis curves, thermomagnetic experiments, progressive acquisition of isothermal remanent magnetization (IRM), and First-Order Reversal Curves (FORCs). Overall, our observations, including low to medium unblocking temperatures, isothermal remanent magnetization to 1 Tesla, and the abundance of wasp-waist hysteresis loops, strongly suggest the presence of Ti-rich titanomagnetites as the main remanence carriers in both lava flows and lapilli, in addition to some hematite as well. Whereas the former has been directly seen (SEM), hematite is elusive with nonmagnetic-based methods. Rock magnetic data, on a Day plot, also reveal that the magnetic grain size tends to be larger in the lava flows than in the lapilli.

**Keywords:** rock magnetism; Canary Islands; recent volcanic eruption; lapilli; basaltic lava flow

## 1. Introduction

The recent volcanic activity of the Cumbre Vieja, in the island of La Palma (Canary Islands, Spain), ejected more than 200 million cubic meters of lava flows and spread over 10 million cubic meters of tephra on the western foothills of the volcano [1]. Volcanic activity lasted a total of 85 days, the longest eruptive process in the island since records for such events began. Previous historic eruptions occurred in 1971 (22 days), 1949 (37 days), and 1712 (56 days) [2], revealing that La Palma is the most active volcanic island in the Canaries. This very recent volcanic activity allows specific studies to be carried out aiming to better understand lava properties, emplacement, lapilli ejection, and deposition in the area, among others. In addition, it furnishes a unique opportunity for rock and paleomagnetism, to better understand how rocks become magnetized during cooling and to investigate to what extent they accurately record both geomagnetic field intensity and magnitude (e.g., [3–12]). A prerequisite for such studies is the characterization of the rock magnetic carriers, i.e., composition, and grain size of the ferromagnetic (s.l.) fraction, which is the main goal of the present study.

During the volcanic eruption, we performed a preliminary, initial sampling of both lava flows and lapilli, to gain knowledge on the rock magnetic properties of such materials. Aerial lavas differ from submarine, oceanic ridge basalts, in that (1) deuteric oxidation is more common due to oxygen fugacity and (2) maghemitization is slow because there is less external water [13]. In addition, volcanic ashes cool down faster than aerial lava flows, a fact that may have implications in the magnetic mineralogy of such materials. Therefore, rock magnetic data from these fresh, aerial lava flows and ashes will eventually

help us better understand how the rock magnetism of aerial volcanic material differs from submarine flows and its implications on paleointensity and secular variation studies. Collecting samples in real time provides direct information about emplacement conditions of the volcanic products and also minimizes the aerial alteration of such deposits, a rare opportunity otherwise. Here, we present an initial report, based on a pilot set of samples (Table 1), while a wider sampling can be conducted when the overall access and conditions allow widespread fieldwork in the area. Accordingly, the rock magnetic data obtained in the present study can provide very useful information to develop specific future paleomagnetic work in this area.

**Table 1.** Sample locations.

| Location | Samples Code | Sample Type | Latitude (°N) | Longitude (°W) |
|---|---|---|---|---|
| Montaña del Cogote | COG1, COG2 | Lava Flow | 28.61131 | 17.89928 |
| La Mariposa | MAR | Lava Flow | 28.60818 | 17.90714 |
| Montaña del Cogote | MC | Lapilli | 28.61013 | 17.88551 |
| Morro de los judíos | MJ | Lapilli | 28.59070 | 17.90661 |
| Finca el Aljibe | FA | Lapilli | 28.63973 | 17.84939 |
| Cabeza de Vaca | CV | Lapilli | 28.62364 | 17.87100 |

## 2. Geological Setting and Microscopic Observations

La Palma and El Hierro are the westernmost islands in the Canary Islands, an archipelago on the African Plate thought to have formed above a mantle plume as the plate moved from WSW to ENE over a hotspot ([14] and references therein). La Palma Island is characterized by a N–S-trending ridge, which is an active volcanic zone [15] (Figure 1). Traditionally (e.g., [16,17], [18] and references therein), the island has been divided into three main units, including a basal complex (3–4 Ma); a volcanic series of 2–0.7 Ma, including the Taburiente shield volcano and the recent Cumbre Vieja series, 0.7 Ma onward. The Cumbre Vieja is a N–S, 25 km long ridge that makes up the southern half of the island, and it is thought to be an active rift zone.

From a compositional point of view, the Cumbre Vieja lava flows include basanite and tephrite to phonolite, with clinopyroxene, olivine, and amphibole as the main phenocrystals in the mafic matrix (e.g., [18]). Our study is focused on the 2021 lava flows, and the composition in detail is still under study. Altogether, our preliminary results based on thin-section observations (Figure 2) and XRD results (Figure 3 and Table 2) reveal the presence of phenocrystals of both clinopyroxene and olivine, in a matrix with abundant plagioclase. Both olivine and pyroxenes have idiomorphic or sub-idiomorphic habits, whereas plagioclase occurs as a microlith in the matrix (Figure 2). Ref. [19] documented the first mineralogical and geochemical analysis for both lava and tephra (erupted in September 2021), reporting that olivine is forsterite (78–80) and the dominant plagioclase has the composition of labradorite. Overall, the mineralogy shows a composition of basanite-tephrite for the 2021 Cumbre Vieja lava flows [19]. In older basanites from Cumbre Vieja, clinopyroxene also occurs as diopside, and Ti-magnetite is present in all rock types in small amounts (<2 vol. %) [18].

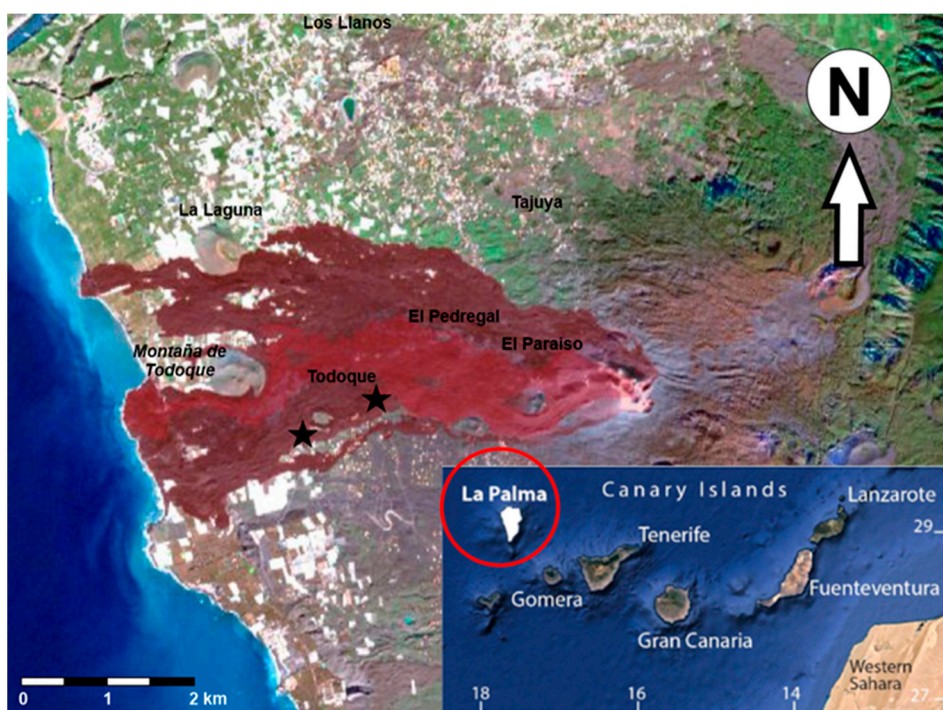

**Figure 1.** Location of Cumbre Vieja within the Canary Islands, including lava-flow sample locations (stars). Lava and lapilli sample location can be found in Table 1.

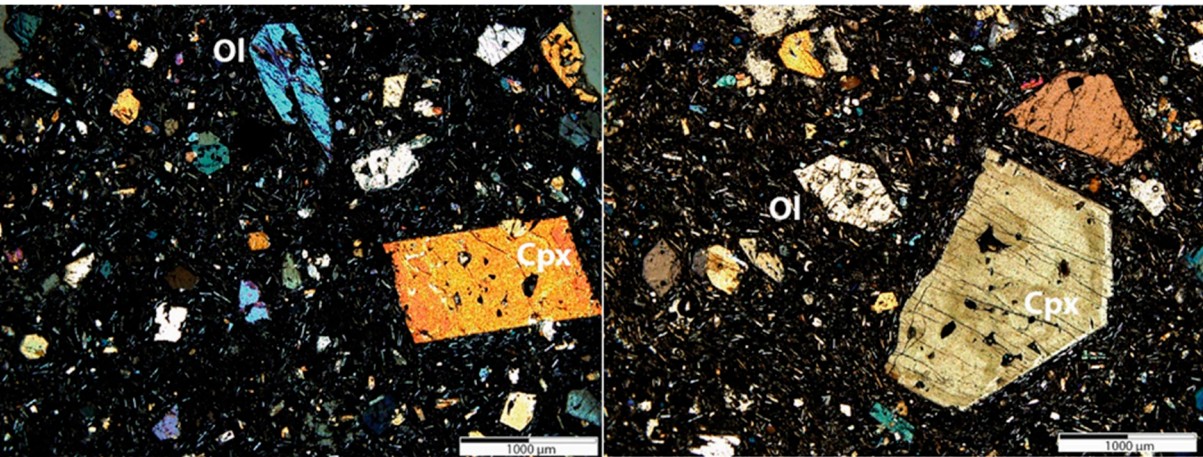

**Figure 2.** Photomicrographs under plane-polarized light illustrating lava flows formed in November, 2021 from Cumbre Vieja, La Palma.

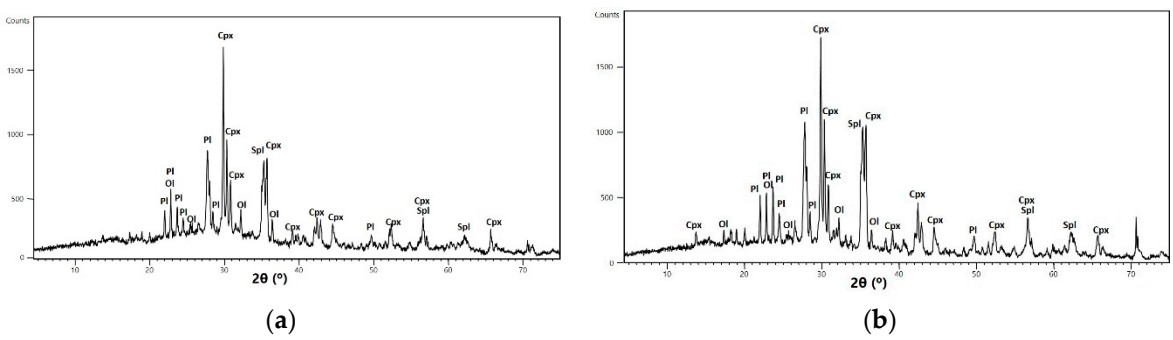

**Figure 3.** X-ray diffraction pattern of lapilli (**a**) (LP-FA) and lava-flow (MARIP) (**b**) samples. Mineral group abbreviations: Cpx = clinopyroxene, Ol = olivine, Pl = plagioclase, and Spl = spinel.



**Table 2.** Semi-quantitative abundance estimations: xxx = very abundant (>40%); xx = major (10–40%); x = minor (<10%).

| Sample | Plagioclase | Clinopyroxene | Olivine | Spinel |
|--------|-------------|---------------|---------|--------|
| LP-COG2 | xxx | xx | xx | x |
| LP-FA | xx | xx | xx | x |
| LP-MARIP | xxx | xx | xx | x |
| LP-MJ | xx | xx | xx | x |

## 3. Samples and Methods

Three lava-flow, unoriented hand samples were collected near Todoque (see Table 1 for sample location). Such a lava flow was emplaced in October 2021. Lapilli and ashes were collected in a number of spots around the volcano (Table 1) in order to cover a wide range of travel distance and grain size.

Laboratory measurements included low-field bulk magnetic susceptibility at two frequencies (976 and 15,616 Hz), progressive isothermal remanent magnetization (IRM) acquisition and demagnetization, progressive alternating field (AF) and thermal demagnetization (TH), low- and high-temperature susceptibility vs. temperature thermomagnetic curves, hysteresis loops, and First-Order Reversal Curves (FORCs). Laboratory measurements were carried out in two paleomagnetic laboratories, including the Archaeomagnetism Laboratory (CENIEH) and the Paleomagnetism Laboratory (University of Burgos). The paleomagnetic experiments carried out at the Archeomagnetism Laboratory of the Geochronology Facilities at CENIEH included the measurement of the natural remanent magnetization (NRM). All remanence measurements (NRM, IRM) were performed in a 3-axis SQUID magnetometer (755 SRM, 2G Enterprises) housed in a ~9 m$^3$ shielded space of Helmholtz coils (residual field < 3000 nT). The cryogenic magnetometer has a built-in system of three degaussing coils for alternating field demagnetization to 170 mT. Thermal demagnetization was performed on a TD48 furnace (ASC) placed in the Helmholtz cage. Rock magnetic experiments carried out at CENIEH also included the measurement of hysteresis loops, and FORC diagrams that were obtained on a MicroMag 3900 vibrating magnetometer (VSM—Princeton Measurements Corp., Princeton, NJ, USA) using the original software and FORCinel software [20]. Bulk, low-field magnetic susceptibility was measured with a MFK1-FA (AGICO), able to measure at three different frequencies. The initial progressive acquisition of isothermal remanent magnetization (IRM) was performed with an ASC pulse magnetizer to 1 Tesla.

Lapilli samples were placed and measured in small gel capsules for the measurements in the Micromag VSM, whereas plastic boxes (ca 1.7 cc) were prepared for the NRM measurements. Lava blocks were cut into small pieces, of less than 8 cm$^3$, to have NRM intensity values low enough to prevent saturation of the cryogenic magnetometer.

Further rock magnetic experiments were performed at the Paleomagnetic Laboratory of the University of Burgos and included thermomagnetic curves and progressive IRM acquisition to higher maximum fields (2.58 T). Temperature-dependent susceptibility curves (k-T) were measured with a Kappabridge KLY-4 device connected to a CS3 furnace for high-temperature (40–700 °C) and a CSL cryostat for low-temperature measurements (−195–0 °C). Experiments were performed in an applied field $H_{app}$ = 300 A/m on powdered lapilli and lava-flow specimens at a rate of 11 °C/min. High-temperature curves of some specimens were recorded in air while others were measured in an argon atmosphere, in order to avoid or at least lessen possible mineral oxidations. Curie points in κ-T curves were determined by taking inflection points in the decreasing branch of the susceptibility curves [21].

Additional progressive IRM acquisition curves up to 2.58 T were carried out with an ASC pulse magnetizer with three different coils to apply increasingly higher fields. For these experiments, approximately 200 mg of either volcanic ash samples or lava-flow specimens broken in a mortar to small pieces were introduced in 1 cm diameter boron-silicate vials

and consolidated with sodium silicate. Remanence was measured with a superconducting 2G cryogenic magnetometer, which includes a degausser for in-line AF demagnetization.

## 4. Results

### *4.1. Bulk Susceptibility*

We measured the low-field, bulk magnetic susceptibility at two frequencies to test the possible presence of superparamagnetism (SP) in the samples. It is generally accepted that the frequency-dependent magnetic susceptibility percentage reflects the relative significance of the SP/SSD (stable single domain) fraction in the total (bulk) magnetic signal [22]. Our results show only slight changes between the two used frequencies, suggesting that the presence of SP particles is very low (Figure 4, Table 3). Few lapilli samples show a more pronounced change, coincident with the lowest susceptibility measurements. We interpret such variation as due to the small amount of measured material.

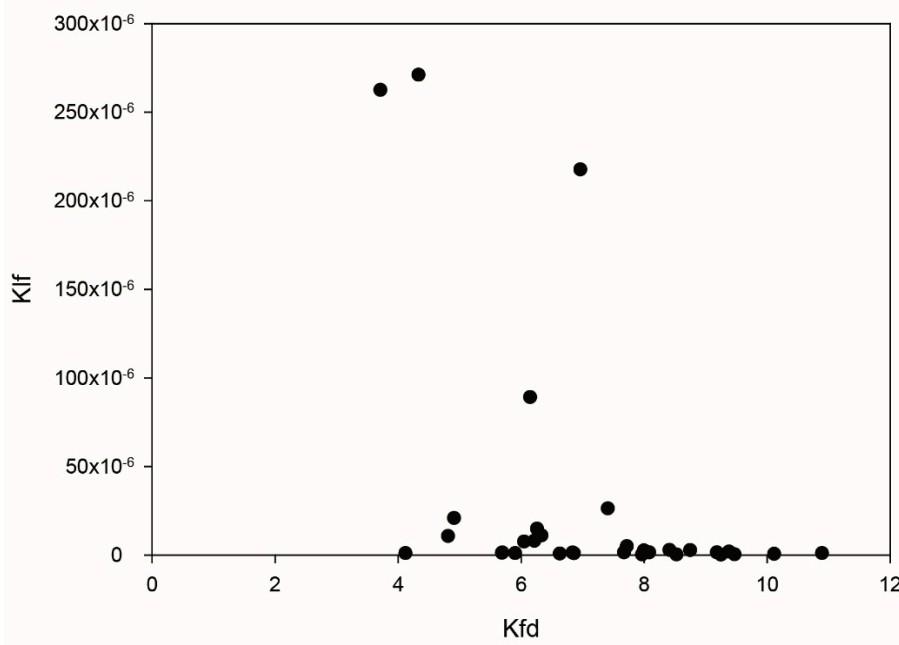

**Figure 4.** Frequency dependent susceptibility (Kfd) as a function of low-frequency susceptibility (Klf). We used Kfd as defined by [22]; Kfd = 100 × ((Klf − Khf)/Klf). Relatively high values of Kfd (>10%) would suggest the concentration of SP particles.

**Table 3.** Rock magnetic measurements. Sample: Sample name; Type: Rock type; $k_b$: Bulk susceptibility (mass-normalized); $k_{LF}$ and $k_{HF}$: low-frequency and high-frequency susceptibility, respectively (volume-normalized); MS: Saturation magnetization; $M_{RS}$: Saturation remanence; $B_C$: Coercivity; $B_{CR}$: Remanent coercivity; SIRM: Saturation isothermal remanent magnetization (at an applied field of 2.8 T).

| Sample | Type | $k_b$ ($10^{-6}$ m³/kg) | $k_{LF}$ ($10^{-6}$) | $k_{HF}$ ($10^{-6}$) | $M_S$ ($10^{-5}$ Am²/kg) | $M_{RS}$ ($10^{-6}$ Am²/kg) | $B_C$ (mT) | $B_{CR}$ (mT) | SIRM (Am²/kg) |
|---|---|---|---|---|---|---|---|---|---|
| FA | Lapilli | 3.65 | 0.60 | 0.56 | 1.75 | 3.54 | 11.9 | 53.0 | 0.16 |
| MC | Lapilli | 2.16 | 0.36 | 0.35 | 0.48 | 0.37 | 3.7 | 18.1 | 0.06 |
| CV | Lapilli | 2.27 | 0.42 | 0.42 | 1.29 | 2.76 | 12.1 | 43.0 | 0.14 |
| MJ | Lapilli | 3.45 | 0.29 | 0.30 | 0.61 | 1.84 | 17.6 | 54.9 | 0.21 |
| COG1 | Lava-flow | 6.33 | 42.59 | 56.22 | 5.70 | 12.20 | 10.4 | 25.0 | 0.41 |
| COG2 | Lava-flow | 13.28 | 23.71 | 22.05 | 5.73 | 11.19 | 10.8 | 36.0 | 0.18 |
| MAR | Lava-flow | 8.12 | 3.02 | 2.81 | 5.32 | 5.89 | 4.9 | 15.3 | 0.07 |

*4.2. Thermomagnetic Curves*

Low-temperature susceptibility vs. temperature curves were measured on five specimens from three different lava-flow samples and two specimens from two different lapilli samples. High-temperature susceptibility vs. temperature curves were measured on five specimens from the same three lava-flow samples and four specimens belonging to three different lapilli samples, the same ones used for low-temperature curves and an additional one. Moreover, in two cases (one lapilli and one lava sample), high-temperature experiments were repeated after having recorded a first heating–cooling run without removing samples from the sample holder. Most high-temperature measurements were performed in air, but some were also in argon.

In low-temperature experiments on lapilli samples, a smooth and marked increase is observed in susceptibility values, the latter showing a strong dependence on temperature (Figure 5C). This behavior is characteristic of Ti-rich titanomagnetites ($Fe_{3-x}Ti_xO_4$ with $x > 0.5$) [23]. In titanomagnetites with $x > 0.04$, the Verwey transition is not detected [24,25], although isotropic points are observed in naturally occurring titanomagnetites with $x < 0.6$ [26–30]. In fact, in lapilli specimen FA, a faint inflection point can be observed at approximately $-110\ °C$ (163 K) (Figure 5C).

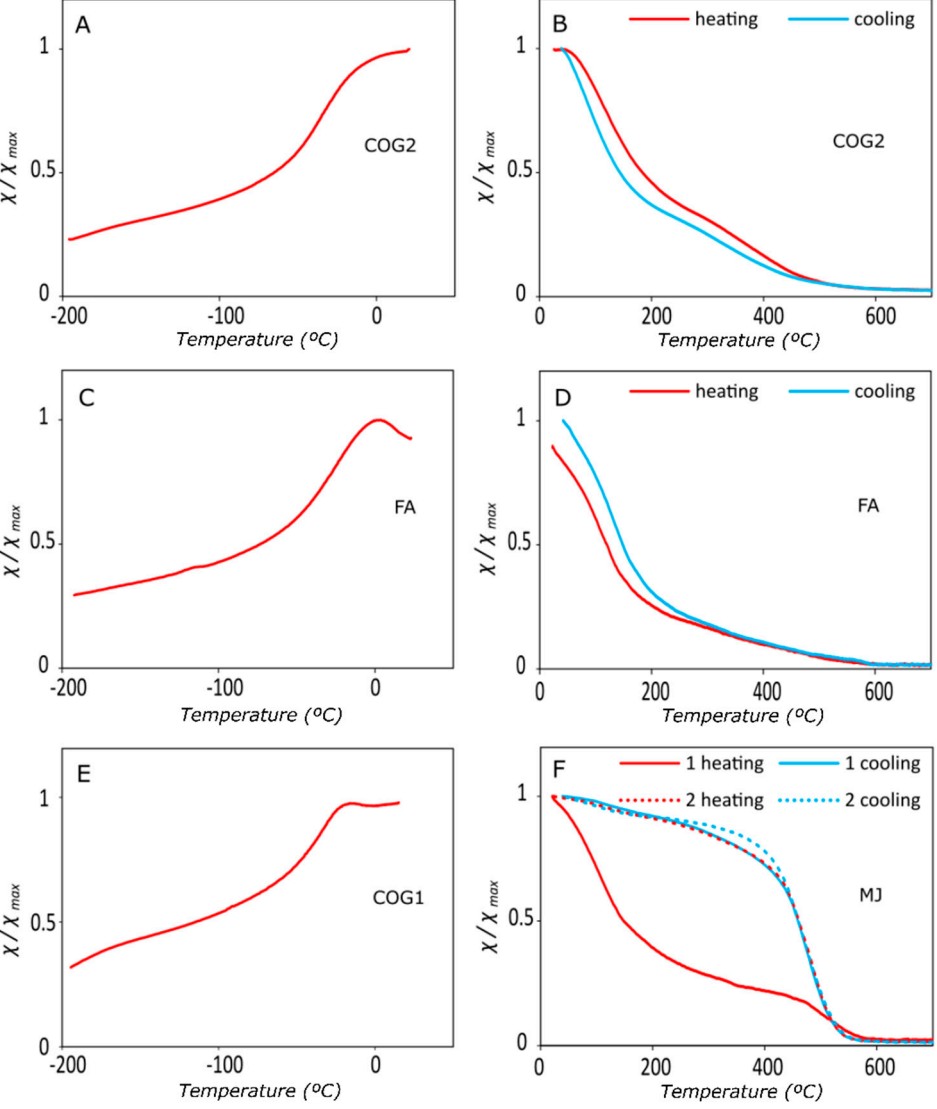

**Figure 5.** Susceptibility vs. temperature ($\chi$-T) thermomagnetic curves for lapilli (FA, MJ) and lava-flow (COG2, COG1) samples. (**A**) shows low-temperature curve for sample COG2. (**B**) shows high-temperature

curve (heating and cooling in argon) for sample COG2. (**C**) shows low-temperature curve for sample FA. (**D**) shows high-temperature curves (heating and cooling in argon) for sample FA. (**E**) shows the Morin transition observed in the low-temperature curve for sample COG1. (**F**) shows the superposition of the high-temperature curve for sample MJ-1 (heating and cooling in air, curves in solid line) and the repetition of the heating process once the measurement was finished (MJ-2, heating and cooling curves represented in dotted line).

High-temperature experiments on three lapilli samples yield similar curves regardless of the atmosphere used, displaying a virtually reversible behavior with a strong low-Curie-temperature phase ($T_C \approx 100$ °C, Figure 5D) and a continuous decrease over the intermediate-temperature interval. These observations point to Ti-rich titanomagnetite phases as the main remanence carrier. A Curie temperature around 100 °C would correspond to a titanomagnetite with x = 0.7 [13], but the continuous susceptibility decrease observed in high-temperature thermomagnetic curves over the intermediate-temperature interval points to the presence of a range of compositions and Curie temperatures. A modest newly created magnetite contribution appears in the cooling curve in two of these thermomagnetic curves. Only lapilli sample MJ shows a somewhat different behavior. In the heating curve, the low-$T_C$ phase ($T_C = 112$ °C) is still present, but instead of a continuous decrease in the intermediate-temperature range as observed in other lapilli samples, a near-magnetite phase ($T_C = 517$ °C) is recognized in the heating curve of this sample. Only the latter is still present in the cooling curve, but with a much stronger contribution (Figure 5F). A second heating–cooling run of the same specimen in air yields a single phase and fully reversible curve (Figure 5F).

As with lapilli samples, two lava-flow specimens from blocks COG2 and MAR display a smooth but strong increase in susceptibility in low-temperature experiments (Figure 5A). Moreover, low-temperature curves were measured also on three different specimens from block COG1, which showed, however, a somewhat different behavior, with a vague inflection around −150 to −140 °C (123 to 133 K) in all three cases together with a clear inflection characterized by a susceptibility maximum around −14 °C (259 K). A slight susceptibility decrease followed this up to −0.5 °C (273 K) with a slight susceptibility increase up to room temperature (Figure 5E). This feature is only observed in two curves and is probably related to the Morin transition [31]. However, the possible transition observed in our samples would not be simple, as it shows a sawtooth shape in a range of some 20 °C between approximately −21 and 0.5 °C. A similar behavior was observed by\ [32] in some lateritic soils from Sulawesi.

Identification of this transition is diagnostic of hematite. At temperatures above the Morin transition temperature $T_M$, a slight "spin canting" leads to weak ferromagnetism in the basal plane, perpendicular to the hexagonal c-axis, while below $T_M$, remanence only arises from defects in the crystal structure. Although $T_M \approx -11$ °C (262 K) for pure stoichiometric hematite at 1 atm [33], the exact value of $T_M$ depends on a number of variables such as grain size, lattice defects, cation substitution, pressure, or applied field [34].

High-temperature curves of three specimens were heated in air. The curves show similar features and remarkable differences. All three display a low-Curie-temperature phase ($T_C \approx 80$ to 200 °C), which is better developed in one of the specimens (COG2), weaker in another (COG1), and mostly hidden by a strong paramagnetic contribution in the third one (MAR). All three specimens also contain a certain amount of intermediate-Curie-temperature phases, which appear to be stronger in the specimen showing the weaker low-$T_C$ contribution. These intermediate-$T_C$ phases do not show a clear Curie point but rather a continuous decrease over a large temperature interval, pointing toward a coexistence of different grain sizes and/or compositions, possibly implying titanomagnetites with varying amounts of Ti. Cooling curves of these samples show an increase in the intermediate-$T_C$ contribution and a decrease or even loss in the low-$T_C$ phase in specimen COG1. Curie temperature values, however, do not change significantly. A second heating–cooling run in sample COG1 in air produces a better-defined Curie point of the intermediate phase,

showing a moderate increase to approximately 500 °C. Repetition of the experiments on sister specimens in argon produced in all cases virtually reversible curves (Figure 5B). While heating curves from COG2 and MAR show similar features to the curves measured in air, two specimens belonging to sample COG1 display a blurring of the low- and intermediate-$T_C$ phases over a wide temperature range without any clear Curie point.

The overall analysis of low- and high-temperature thermomagnetic curves of lava samples points to titanomagnetites as the main remanence carrier. In this case, however, together with a low-$T_C$, high-Ti phase, a stronger contribution than in lapilli samples of an intermediate-$T_C$ phase is observed. It is noteworthy that despite observing a Morin transition in samples from lava COG1, no presence of hematite is observed in high-temperature curves, probably because the weak susceptibility of hematite is hidden by the more substantial contribution of the titanomagnetite phases.

### 4.3. IRM Acquisition and Component Analysis

Isothermal remanent magnetization (IRM) acquisition curves are extremely useful as the first line of defense for characterizing the ferromagnetic composition in rocks, allowing the detection of the presence of phases with different coercivities (e.g., [35,36]). Initially, samples can be considered as demagnetized, because for IRM experiments, they have been powdered and subsequently consolidated. At first, we measured the progressive IRM acquisition of a set of specimens to a maximum DC field of 1 Tesla, revealing that all samples, lapilli and lava flows, experience a sharp increase in magnetization up to around 0.20 T and a more plateau-like behavior in all samples above 0.45 T. We notice a slight difference between the lava-flow and lapilli samples, in that the former reaches saturation abruptly and the latter show a more progressive acquisition of magnetization between 0.1 and 0.4 T before reaching saturation. Therefore, we carried out several more detailed IRM acquisition experiments to higher DC fields of 1.84 and 2.58 T. It is interesting to note that while lava-flow samples reach 80% of their saturation value at about 50–70 mT, lapilli specimens attain 80% saturation at a higher field of 130–140 mT. (Figure 6). Such contrasting behavior probably reveals that, whereas magnetite (*sensu lato*) is the main carrier in both materials, lapilli have a broader spectrum of grain sizes and, therefore, a more progressive acquisition of saturation. We return to this point after describing further rock magnetic evidence. Saturation (SIRM, Table 3) is attained in most cases between 0.7 and 1.1 T, although, in lava-flow sample COG1, a strong (1.84 T) field is necessary to reach saturation. IRM acquisition curves of two of the three analyzed lavas show the presence of a weak high-coercivity fraction, with SIRM being about 0.75% higher than IRM at 1 T.

Progressive IRM acquisition was performed on three lava-flow samples and four lapilli samples for component analysis. The unmixing coercivity data were performed with the MAX UnMix web application [37], obtaining the following results (Table 4).

At least two low-coercivity components (component 1 and component 2) have been differentiated in all the studied samples, both in a relatively high proportion. In addition, it is possible to distinguish a high-coercivity component (component 3) in most of the samples in very low proportion (Figure 7). Component 3 has been recognized in all samples except for sample COG1 (lava-flow), although in a very small amount (Figure 7A). It should be pointed out that while adding a third component might help for a better curve adjustment, it does not necessarily indicate its real presence in the samples. Still, the presence of hematite in several samples (i.e., MC, COG2, CV, and MAR) cannot be ruled out.

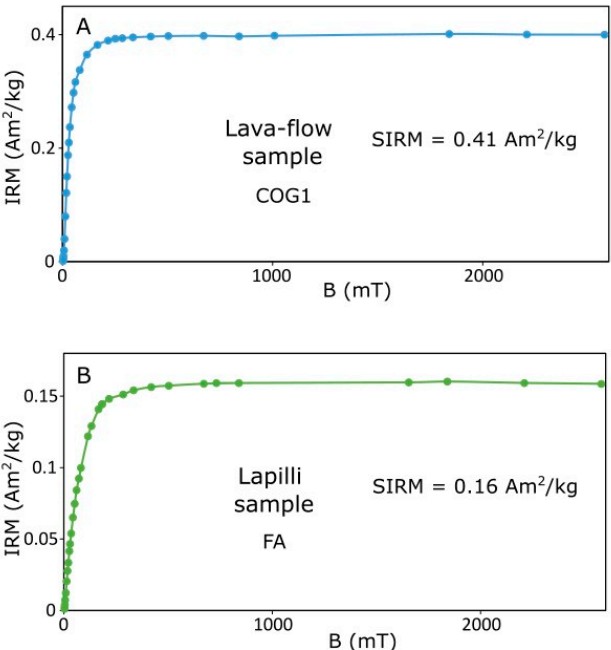

**Figure 6.** Progressive IRM acquisition curves for representative samples. (**A**) COG1 (lava-flow); (**B**) FA (lapilli).

**Table 4.** Results of model fitting for seven selected samples from lava flows and lapilli using the MAX UnMix software [37]. $B_{1/2}$: median acquisition field; DP: dispersion parameter; S: parameter-describing skewness; EC: relative extrapolated contribution of each component to the total measured magnetization; IRM: isothermal remanent magnetization of each component. Units: $B_{1/2}$ in mT; IRM in $10^{-3}$ Am$^2$/kg; DP, S, and EC are dimensionless.

| | | | | | | | | | | | | | | | | | |
|---|---|---|---|---|---|---|---|---|---|---|---|---|---|---|---|---|---|
| | | | | | | | **IRM Acquisition** | | | | | | | | | | |
| **Sample** | | **Component 1** | | | | | | **Component 2** | | | | | | **Component 3** | | | |
| **Ref.** | **Type** | **Log ($B_{1/2}$)** | **$B_{1/2}$** | **DP** | **S** | **EC** | **IRM** | **Log ($B_{1/2}$)** | **$B_{1/2}$** | **DP** | **S** | **EC** | **IRM** | **Log ($B_{1/2}$)** | **$B_{1/2}$** | **DP** | **S** | **EC** | **IRM** |
| COG1 | Lava | 1.30 | 20.00 | 0.32 | 0.93 | 0.63 | 260 | 1.79 | 61.58 | 0.34 | 0.95 | 0.37 | 150 | - | - | - | - | - | - |
| FA | Lap. | 1.38 | 24.01 | 0.46 | 0.80 | 0.47 | 76 | 1.90 | 78.80 | 0.28 | 0.68 | 0.46 | 75 | 2.51 | 324.44 | 0.24 | 1.03 | 0.07 | 11.4 |
| MC | Lap. | 1.13 | 13.59 | 0.43 | 0.74 | 0.46 | 275 | 1.80 | 62.34 | 0.33 | 0.72 | 0.50 | 300 | 2.60 | 401.08 | 0.19 | 1.09 | 0.04 | 24.3 |
| CV | Lap. | 1.36 | 23.10 | 0.49 | 0.90 | 0.48 | 66 | 1.90 | 78.52 | 0.31 | 0.73 | 0.49 | 68 | 2.64 | 436.37 | 0.16 | 1.08 | 0.04 | 5.1 |
| MJ | Lap. | 1.31 | 20.32 | 0.39 | 0.91 | 0.39 | 84 | 1.96 | 90.62 | 0.31 | 1.03 | 0.57 | 124 | 2.30 | 200.60 | 0.15 | 1.02 | 0.04 | 7.8 |
| COG2 | Lava | 1.19 | 15.64 | 0.36 | 1.07 | 0.54 | 97 | 1.87 | 74.31 | 0.28 | 0.72 | 0.43 | 77 | 2.60 | 400.53 | 0.14 | 1.08 | 0.03 | 4.6 |
| MAR | Lava | 0.99 | 9.87 | 0.35 | 1.11 | 0.47 | 31 | 1.68 | 48.21 | 0.33 | 0.97 | 0.43 | 29 | 2.77 | 592.47 | 0.48 | 1.12 | 0.10 | 6.7 |

The lowest-coercivity component (component 1) has a median acquisition field ($B_{1/2}$) of ~18 mT on average, and sample MAR (lava-flow) shows the minimum value (~10 mT) (Figure 7C). It is possible to distinguish a higher proportion of component 1 in lava flows (between 63% and 47%) than in lapilli samples (45% on average). The dispersion parameter DP is significantly higher in lapilli samples than in lava flows, where this first component seems to be better defined. Component 2 has a slightly higher coercivity, with $B_{1/2}$ values of ~70 mT on average. Sample MAR represents, again, an exception, showing a significantly lower (48 mT) $B_{1/2}$ value. In contrast to component 1, its proportion is higher in lapilli (50%) than in lava flows (41%). As stated above, a high-coercivity component (component 3) has been recognized in lapilli samples FA, MC, CV, and MJ, and lava-flow samples COG2 and MAR; however, its presence is uncertain due to the poor accuracy at the end of the IRM spectrum. $B_{1/2}$ values for this third component deviate between 200 mT and 600 mT, with a very small DP parameter, except for MAR. Its proportion is, in all cases, less than 10%.

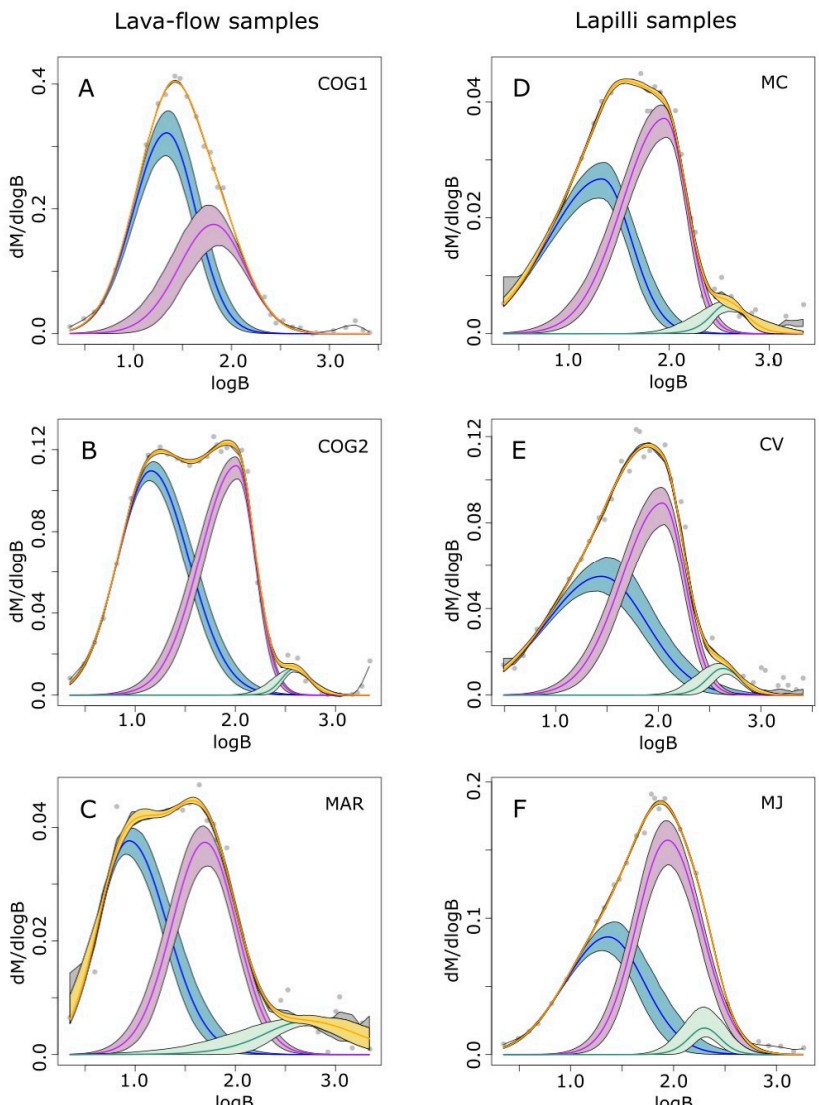

**Figure 7.** Unmixing diagrams ([37], see text) for three lava-flow samples (**A–C**) and three lapilli samples (**D–F**). The yellow line represents the final model adjustment. Adjustment for component 1 is represented in blue, for component 2 in purple, and for component 3 in green. Units: B in mT and M in Am$^2$/kg.

Coercivity values for components 1 and 2 would represent titanomagnetite phases with different compositions and/or grain sizes. The grain size of component 1 would be significantly larger than that of component 2. High coercivity values for component 3 (~400 mT) may indicate the presence of single-domain hematite [13]. In general, all samples seem to contain two different populations of titanomagnetites of different compositions and/or grain sizes, which may also contain a small proportion of single-domain hematite. Lava flows show a higher proportion in larger-grain-size titanomagnetites than lapilli samples. Lava-flow MAR presents lower $B_{1/2}$ values for both component 1 and 2, meaning titanomagnetites of larger grain size, and a higher component 3 (hematite) proportion.

### 4.4. Hysteresis Curves

Hysteresis loops were performed to a field of 1 Tesla. Hysteresis parameters saturation remanence ($M_{rs}$), saturation magnetization ($M_s$), remanent coercivity ($B_{cr}$), and coercivity ($B_c$) are shown in Table 3. As IRM experiments have already revealed, most samples nearly reach saturation around 0.40 mT, whereas some show a paramagnetic contribution above

that field. Overall, both ferromagnetic and paramagnetic fractions are apparent, although most curves do not need to be corrected for such a contribution.

Based on the shape of the hysteresis curves, we observe both "standard" curves, whereas numerous loops are narrower, especially at the middle of the curves, and result in constricted shapes, or the so-called wasp-waist loops [38,39] (Figure 8). "Wasp waist" can result from either a mixture of two populations of magnetic grains with distinct coercivity or a population close to the SP/SD threshold [39]. Either way, and because we are studying a single composition material (basanite), the use of the $M_{rs}/M_s$ and $B_{cr}/B_c$ ratios is adequate (see discussion in [40]) and, therefore, the Day diagram can furnish information on the magnetic grain size of the studied volcanic material. We noticed that, whereas lava-flow sample ratios fall within the area where other aerial flows are found (e.g., Azores, Mt. St. Helens, and Vesuvius; see Figure 8), lapilli sample ratios present higher $B_{cr}/B_c$ ratios. Such a distribution possibly reveals a higher content in MD grain fraction, an observation that is coherent with the thermomagnetic curves as described before.

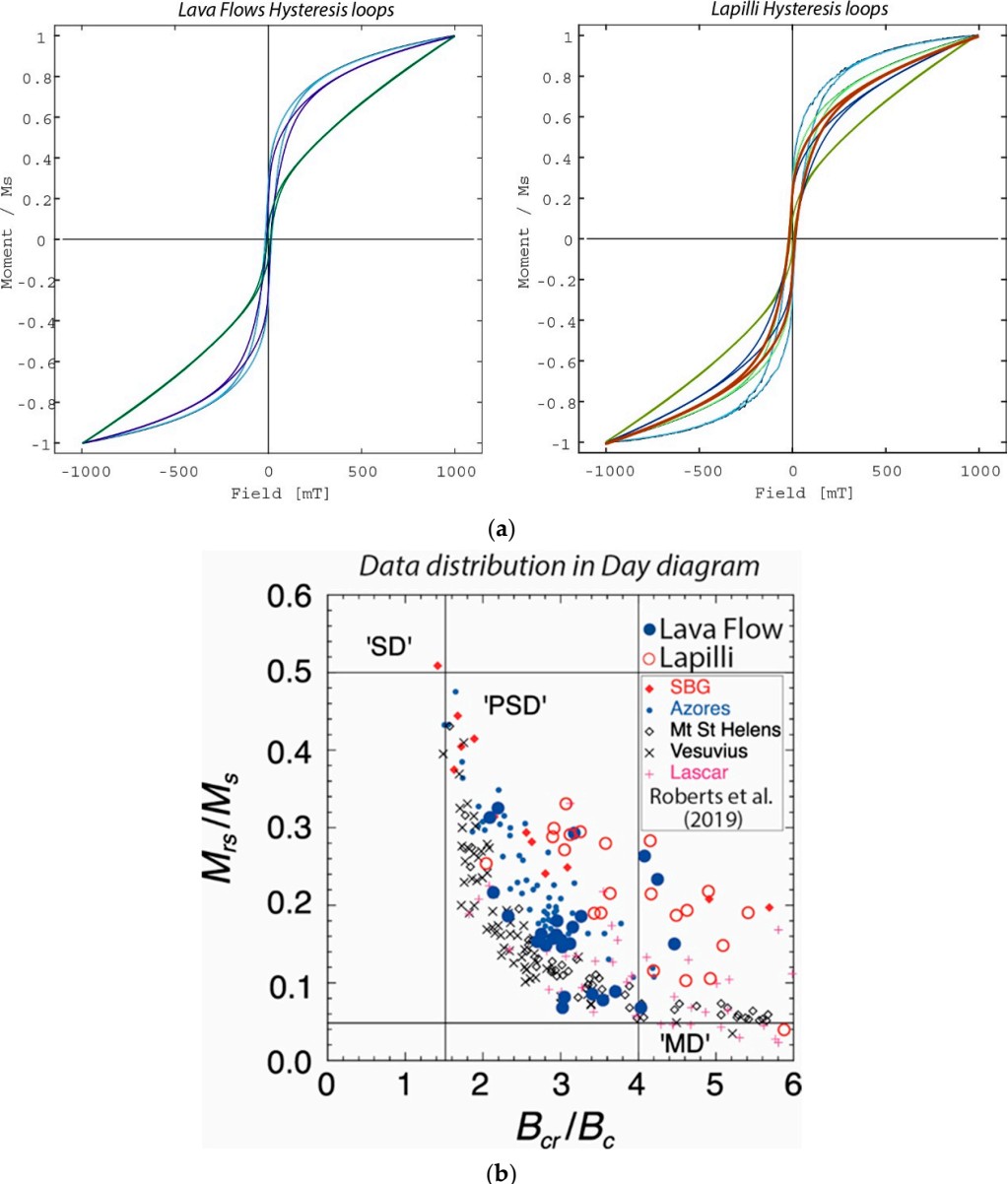

**Figure 8.** (**a**) Hysteresis loops for representative samples and (**b**) Day diagram for different basaltic rocks (modified from [40]). See text for discussion.

Last, First-Order Reversal Curve (FORC) diagrams (Figure 9, Table 5) show that, generally, the inner contours converge around a central peak, while there are no signs for a pronounced "central ridge" for most samples. The distribution has a prominent peak centered about the origin of the diagram, a manifestation of SD particles [41]. The sample in Figure 9A shows stronger evidence for such SD particles, whereas there is also evidence for low-coercivity coarse MD particles. The elongation of the x-axes could result from a high-coercivity maghemite/hematite component.

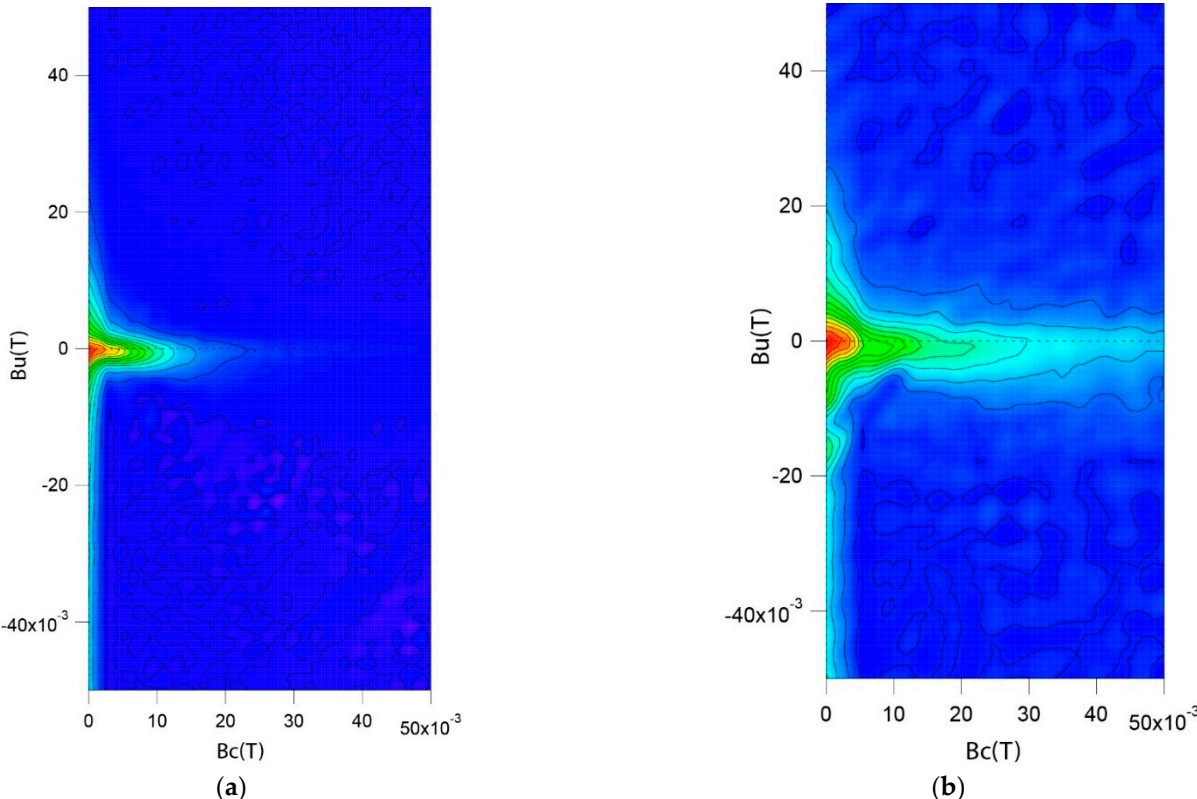

**Figure 9.** First-Order Reversal Curves (FORCs) for representative samples. (**a**) Lava flow (COG2); (**b**) lapilli (MJ).

**Table 5.** FORC parameters.

| Parameter | Value |
| --- | --- |
| Saturating field | 500 mT |
| Hu | ±50 mT |
| Averaging time | 0.15 s |
| Hb1 | −500 mT |
| Hb2 | 500 mT |
| Number of FORCs | 300 |
| Moment Range | 0.5 T |
| Field increment | 1.60 mT |
| Pause at saturating field | 1 s |
| Slew rate limit | 1 T/s |
| HC (min) | 0 T |
| HC (max) | 120 mT |

## 5. Final Remarks

Low to medium unblocking temperatures as revealed by thermomagnetic curves indicate the presence of Ti-rich magnetite as the main carrier of remanence in both lava flows and lapilli. The presence of Ti-magnetite is in agreement with the results by scanning

electron microscope (QEMSCAN®) analysis [19], which shows such oxides both as discrete grains and in the groundmass with a concentration of TiO$_2$ between 3 and 5 wt %. Hysteresis data, particularly the abundance of wasp-waisted loops in both lava flows and lapilli, suggest a mixture of coercivity components (e.g., [38,39]). The abundance of wasp-waisted loops combined with the isothermal remanent magnetization component analysis indicates multiple coercivity fractions that are compatible with the presence of magnetite and small amounts of hematite. It is worth noting that the presence of hematite was not seen by the scanning electron microscope [19]; however, the end member ilmenite (1.95 modal %) was detected either as equant or as microlithic crystals.

Rock magnetic data also reveal that, overall, the magnetic grain size tends to be larger in the lapilli than in lava flows, an observation that needs to be further explored. Last, both thermomagnetic curves and wasp waist hysteresis loops are compatible with small amounts of hematite, in addition to the prevailing magnetite.

As rock magnetic measurements show that the lava-flow samples contain a relatively wide range of coercivities, at least a part of the ferromagnetic (s.l.) phases in these rocks can carry a remanent magnetization able to provide a stable paleomagnetic signal supplying reliable information for secular variation studies.

The very reversible thermomagnetic curves observed in the Cumbre Vieja lava flows indicate that these rocks can be suitable for paleointensity studies. However, the coexistence of a mixture of grain sizes, including MD grains in different amounts, would limit, in certain cases, their use for Thellier-type experiments. Paleointensity experiments considering the specific location of samples in a flow to detect how much the lava-flow cooling rate affects grain sizes would supply new, interesting information. Nevertheless, given the thermal stability of the samples and their relatively high MD content in some cases, successful paleointensity determinations could be performed with the multispecimen method (e.g., [42]).

**Author Contributions:** Conceptualization, J.M.P., M.C.-R. and V.S.; methodology, J.M.P., E.V., M.C.-R. and M.-F.B.; investigation, J.M.P., E.V., M.-F.B. and A.Á.; resources, J.M.P. and M.C.-R.; writing, original draft, J.M.P., E.V. and M.C.-R.; review and editing, J.M.P., E.V., M.C.-R. and M.-F.B.; visualization, J.M.P., E.V. and M.C.-R.; project administration, J.M.P., M.C.-R. and V.S.; funding acquisition, J.M.P., M.C.-R. and V.S. All authors have read and agreed to the published version of the manuscript.

**Funding:** This research was funded by Agencia Estatal de Investigación, Spain, grant no. PID2019-105796GB-00/AEI/10.13039/501100011033.

**Institutional Review Board Statement:** Not applicable.

**Informed Consent Statement:** Not applicable.

**Data Availability Statement:** The data presented in this study are available on request from the corresponding author.

**Acknowledgments:** We thank Antonio González (Cabildo de La Palma) for his invaluable support and help during our visit to the island of La Palma. Carlos Sáiz (CENIEH) is thanked for preparing thin sections. We also wish to thank three anonymous reviewers for their useful and constructive remarks.

**Conflicts of Interest:** The authors declare no conflict of interest.

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
