# Peer review of "Rock Magnetism of Lapilli and Lava Flows from Cumbre Vieja Volcano, 2021 Eruption (La Palma, Canary Islands): Initial Reports"

_geosciences, doi:10.3390/geosciences12070271_

Round 1

Reviewer 1 Report

The rock magnetic studies proposed in this manuscript from aerial lava flows and ashes will provide a better understanding of the rock magnetism of aerial volcanic material and how it differs from submarine flows; also its implications on paleointensity and secular variation studies. Concerning this last item, the authors don´t make any interpretation. Then, they should comment on how the founded results could influence secular variation and paleointensity studies or remove this statement.

The manuscript is understandable, well-organized, and gives us clear explanations of the findings, making it easy to read.

1.       Abstract: I suggest the authors include more information about the results than the measurement techniques.

2.       Samples and Methods: The methodology is arranged in a bit disorderly way. It would be necessary to rearrange it. In particular, the authors mentioned measurements of paleomagnetic experiments performed at the Archeomagnetism Laboratory of the Geochronology Facilities at CENIEH, but they do not describe the results. They should tell about them or remove this information.

3.       I made some recommendations about using English. Please, see the attached PDF for the rest of the comments and corrections.

Author Response

Please, see attachment

Reviewer 2 Report

Minors suggestions:

line 97: Please provide information about the parameters used to measure and analyze the FORC diagrams and the software used to process the FORC measurements.

line 108: Correct spelling is FORC not FORD

lines 111-113: Please clarify this sentence with respect to lines 128-129. Different instruments?

lines 353-354: Please discuss this conclusion with respect to the frequency-dependent magnetic susceptibility results for lapilli samples that do not support the presence of SP particles (lines 143-146).

Author Response

Please, see attachment

Reviewer 3 Report

This contribution is interesting as in provides rock magnetic data on pristine volcanic material (a 2021 lava flow). It can be of general interest and worth publishing. However, a table summarizing the rock magnetic parameter measured is mandatory. This data can be useful for further research! Present mass normalized data of course; Xhi, IRM, S ratio, hysteresis parameters…

For example the magnetic susceptibility chapter gives no data apart Fig.4 that has a major problem : what is the unit?

details:

Fig.4 is not very useful; better plot frequency dependence as a function of Xhi (mass normalized). I have a very hard time to believe that your lava sample value ranges over 4 orders of magnitude on a single flow, so I guess it’s a problem of mass normalization. According to Fig.5 Xhi is around 10-5 m3/kg, while in fig.4 it varies down to 10-7; ??

« High-temperature experiments on three lapilli samples yield rather similar curves, displaying a virtually reversible behavior » Well, I doubt that Fig.5A and B present the cooling curves allowing to demonstrate this.

« For IRM acquisition and demagnetization experiments both volcanic ash and lava flow samples were crushed in a mortar « . Strange choice, you may have modified the coercivity of the pristine grains by the crushing.

Due to lack of table we are unable to check if the proposed Morin transition has been detected high coercivity samples. I am very reluctant in accepting that a Morin transition can be seen on susceptibility curve while the hematite signal is minor in remanence.

« The FORC distribution has a large peak centered about the origin of the diagram, a manifestation of SD particles « Are you sure ? My understanding of FORC is that evidence for SD correspond to a maxima away from the X origin.

« Hysteresis data, in particular the abundance of wasp-waisted loops in  both lava flows and lapilli, are suggestive of a mixture of remanence components « why remanence ? I would say coercivity.

Author Response

Please, see attachment

Round 2

Reviewer 3 Report

The revised version has solved all but one problem I had with the previous version and is now near ready for publication. The last problem is again with fig.4. Thanks to table 3 it is now clear that the amount of ferrimagnetic grains vary only by a factor of 10 (ratio of CG2/MC either on kb or Ms). So the factor of a few 1000 seen on fig. 4 where you say it’s volume normalized data is an artefact of the fact that you normalized to a nominal volume (I guess 10 cc) instead of real volume of the sample. Another way to see the problem : in table 3 the ratio of klf to kb should give the bulk density (i.e. around 2000 kg/m3). That’s far from being the case. If you want to show the actual measurement of the instrument (i.e. without normalizing to sample mass or volume) for the sake of noise evaluation, it’s acceptable, but you have to state it. Again your plot does not allow to evaluate precisely the frequency dependence. Please plot fd as a function of klf instead of khf as a function of klf.